# ‘Exerkines’: A Comprehensive Term for the Factors Produced in Response to Exercise

**DOI:** 10.3390/biomedicines12091975

**Published:** 2024-09-01

**Authors:** Giuseppe Novelli, Giuseppe Calcaterra, Federico Casciani, Sergio Pecorelli, Jawahar L. Mehta

**Affiliations:** 1Department of Biomedicine and Prevention, Tor Vergata University of Rome, 00173 Rome, Italy; cascio94@gmail.com; 2Giovanni Lorenzini Medical Foundation, 20129 Milan, Italy; sergiopecos@gmail.com (S.P.); mehtajl@uams.edu (J.L.M.); 3Giovanni Lorenzini Medical Foundation New York, Woodcliff Lake, NJ 07677, USA; 4Italian Federation of Sports Medicine, 00196 Rome, Italy; 5Postgraduate Medical School of Cardiology, University of Palermo, 90127 Palermo, Italy; peppinocal7@gmail.com; 6School of Medicine, University of Brescia, 25123 Brescia, Italy; 7Department of Medicine (Cardiology), University of Arkansas for Medical Sciences, Little Rock, AR 72205, USA

**Keywords:** exerkines, physical exercise, inflammation, omics, biomolecules

## Abstract

Regular exercise and physical activity are now considered lifestyle factors with positive effects on human health. Physical activity reduces disease burden, protects against the onset of pathologies, and improves the clinical course of disease. Unlike pharmacological therapies, the effects mediated by exercise are not limited to a specific target organ but act in multiple biological systems simultaneously. Despite the substantial health benefits of physical training, the precise molecular signaling processes that lead to structural and functional tissue adaptation remain largely unknown. Only recently, several bioactive molecules have been discovered that are produced following physical exercise. These molecules are collectively called “exerkines”. Exerkines are released from various tissues in response to exercise, and play a crucial role in mediating the beneficial effects of exercise on the body. Major discoveries involving exerkines highlight their diverse functions and health implications, particularly in metabolic regulation, neuroprotection, and muscle adaptation. These molecules, including peptides, nucleic acids, lipids, and microRNAs, act through paracrine, endocrine, and autocrine pathways to exert their effects on various organs and tissues. Exerkines represent a complex network of signaling molecules that mediate the multiple benefits of exercise. Their roles in metabolic regulation, neuroprotection, and muscle adaptation highlight the importance of physical activity in maintaining health and preventing disease.

## 1. Introduction

There is no doubt that regular exercise brings benefits in terms of prolonging life, improving overall health, and preventing disease [1]. Walking for 60 min every day, about 400 min every week, has positive effects on longevity and reducing the risk of cardiovascular disease, neurological disease, cancer, and type 2 diabetes [1,2]. Although the terms ‘exercise’ and ‘physical activity’ are commonly used interchangeably, exercise is typically regarded as intentional physical activity, such as aerobic training, resistance training or high-intensity interval training [3,4]. By contrast, physical activity encompasses exercise as well as usual occupational and/or domestic activity.

However, an important question remains: how does exercise produce positive biological effects on human health? Over the past decade, researchers have begun to decipher different cellular and molecular pathways that are activated throughout the body during exercise; some of these persist even after exercise. On a biochemical level, exercise triggers the secretion of proteins known as myokines from skeletal muscles, such as interleukin-10 and irisin, which have beneficial effects on a host of tissues and reduce the concentration of proinflammatory cytokines, contributing to overall improvements in health [5]. Skeletal muscles represent ~30–40% of the total body’s weight and allow us to perform a wide range of movements and functions. Skeletal muscles are voluntary, meaning you control how and when they work. They are also a reservoir of amino acids stored as protein. Skeletal muscle, with its marked plasticity, is capable of adaptation throughout life in response to a variety of signals including neural activation, mechanical loading, growth factors, and nutritional status [6]. Skeletal muscle proteins are constantly and simultaneously synthesized and degraded. Net protein balance is defined as the difference between skeletal muscle protein synthesis and muscle protein breakdown [7]. The balance between anabolism and catabolism is crucial for muscle mass and functions. Indeed, protein balance does not function as a simplistic binary operation (e.g., synthesis or degradation) but instead as a summation of multiple processes that dynamically operate in an interconnected network [8].

Skeletal muscle exercise burns up energy, especially glucose that would otherwise be stored as fat, which, in excess amounts, increases the risk of cardiovascular disease, type 2 diabetes, and some cancers. Opposite, even short periods of physical inactivity are associated with impaired metabolic homeostasis, manifested as decreased insulin sensitivity and reduced postprandial lipid clearance, loss of muscle mass, and an accumulation of visceral adiposity [9,10]. Exercise impacts various bodily systems, such as the cardiovascular system, improving insulin sensitivity, increasing muscle volume, decreasing body fat, and increasing blood flow to trained muscles, all of which contribute to improved health outcomes [11]. Furthermore, exercise has been shown to improve neurogenesis, increase the expression of neurotrophins such as brain-derived neurotrophic factor (BDNF), promote dendritic remodeling, and stabilize stress responses and inflammatory signaling in the brain, highlighting its significant benefits for mental well-being [12]. However, only in recent years has research been able to provide coherent biochemical and molecular data through a multisystem and multiomics approach. The results of a comprehensive study, the “Molecular Transducers of Physical Activity Consortium” (MoTrPAC), analyzed the transcriptomic, proteomic, metabolomic, and lipidomic profiles of 18 solid tissues from a sex-controlled group of rats (*Rattus norvegicus*) during 8 weeks of resistance training. The investigators also analyzed the phosphoproteome, acetylproteome, ubiquitylproteome, epigenome, and immunome in whole blood and plasma [13]. By mapping dynamic responses to exercise, MoTrPAC generated a molecular map of exercise, providing insight into the intricate molecular transducers that mediate the effects of physical activity on various tissues and organs [14]. By analyzing the molecular basis of physical activity and resilience, the consortium aimed to identify and characterize molecular transducers, including exerkines, actives in both humans and animal models [15]. 

## 2. Exerkines: Mediators of Benefits of Physical Activity

Exerkines, also known as exercise-induced bioactive molecules, have emerged as key mediators of the health benefits associated with physical activity (Table 1). 

These molecules are released from various tissues in response to exercise and play a critical role in promoting general well-being and regulating metabolic homeostasis, thus preventing metabolic diseases [16]. Exerkines have been identified as signaling molecules that exert influence on a wide range of intricate processes in a variety of tissues such as muscle, adipose tissue, pancreas, liver, cardiovascular system, kidney, and bone [17]. 

Several exerkines, including FGF21, IL-6, adiponectin, irisin, apelin, and myonectin, have been identified and studied for their therapeutic potential in the treatment of metabolic and cardiovascular diseases [18,19]. These molecules have been shown to promote cross-talk between organs, mediate endocrine effects that attenuate aspects of metabolic syndrome, such as fatty liver, dysglycemia, insulin resistance, increased adiposity, and exercise intolerance [20]. Exerkines play a crucial role in various physiological processes, including skeletal muscle development and growth, tissue regeneration, and cognitive improvement [21,22,23]. They can be secreted by a variety of cells, acting as autocrine, paracrine, or circulating regulators in response to exercise, and contribute to the systemic effects of physical activity. Furthermore, exercise-induced extracellular vesicles enriched with exerkines have been identified as a new class of molecules that promote systemic beneficial effects [24].

However, the question remains as to the number of exerkines, their source, and the regulation of their expression. Biochemically, exerkines can be hormones, metabolites, peptides, proteins, and nucleic acids; their pleiotropic nature determines the response of various physiological systems to physical exercise (Table 1).

The first exerkine identified was IL-6 as a myokine in 2000 [25]. Since then, numerous exerkines have been characterized in the cardiovascular system, endocrine system, nervous system, immune system, adipose tissue, skeletal muscle, liver, and intestine [4]. IL-6 is a multifunctional cytokine that plays a crucial role in various physiological and pathological conditions across different systems. IL-6 is produced in response to infections, tissue injuries, and stress, contributing to host defense mechanisms through the stimulation of acute-phase responses, haematopoiesis, and immune reactions [26]. IL-6 exerts its effects through binding to its receptor, which is present in various tissues, including the central nervous system [27]. The cytokine IL-6 has pleiotropic effects in different cell types and plays a crucial role in various physiological processes, including inflammation, immune response, and metabolic regulation [28]. Some other major exerkines include Oxylipin 12, 13 diHOME;—this exerkine is involved in metabolic health and systemic metabolism [29]; adiponectin, a lipid hormone that influences metabolic health and insulin sensitivity [16]; brain-derived neurotrophic factor (BDNF), known for its role in promoting neuroprotection and cognitive function [16]; lactate, considered a major myokine and exerkine, promoting beneficial metabolic and anti-inflammatory effects of exercise [30]; irisin, an exerkine produced during muscle contraction that contributes to anti-inflammatory effects and homeostasis [31]; Fibroblast growth factor 21 (FGF21), implicated in obesity, insulin resistance, and type 2 diabetes; fibronectins, exerkines like irisin that have anti-inflammatory properties and play a role in aging and redox-mediated comorbidities [31]. Exerkines have been identified as important regulators of processes such as adipose tissue browning, with specific exerkines like irisin, meteorin-like (METRNL), and FGF-21 playing key roles in systemic metabolic adaptations [20]. Apelin is an exerkine relevant to the metabolic control of type 2 diabetes mellitus [3]. These exerkines, through paracrine, endocrine, and autocrine pathways, mediate the molecular effects of exercise on the entire organism, highlighting their significance in promoting metabolic health and overall well-being [32]. Exerkines have been associated with various health benefits, including promoting angiogenesis, enhancing endothelial cell function, and potentially rescuing cognitive decline [33,34]. These molecules have been linked to the regulation of neurodegenerative diseases and have shown promise in the treatment of conditions like Alzheimer’s disease [35,36]. Interestingly, Mohammad et al. [37] described how voluntary running increased the concentration of an enzyme (BACE1) that limits the overproduction of beta-amyloid precursor protein in ovariectomized mice. Exercise has been shown to have positive effects on various diseases, with specific exercise prescriptions recommended for conditions such as cancer, Parkinson’s disease, and cardiovascular diseases [38,39]. The European Association of Preventive Cardiology has developed tools like the EXPERT system to optimize exercise prescriptions for cardiovascular disease patients, considering factors like exercise tolerance, medications, and adverse events during testing [40]. Exercise therapy is considered an active intervention for the rehabilitation of various diseases [41]. In this context, exerkines constitute a mechanistic link between exercise and its beneficial effects. For example, exerkines have recently been shown to improve neurogenesis and neuroprotection, thereby promoting brain health [42]. In elderly individuals with mild cognitive impairment, acute aerobic or strength exercise have been shown to alter circulating exerkine levels, affecting neurocognitive functions [43]. Furthermore, platelet-derived exerkines like CXCL4/platelet factor 4 have been found to enhance hippocampal neurogenesis and restore cognitive function in aged mice [44]. These findings highlight the potential of exerkines in promoting cognitive health and neurogenesis.

Barres et al., shown that a long time that exercise alters epigenetics and causes short-term changes in DNA methylation and gene expression in muscle tissue that may have implications for type 2 diabetes [45]. In 2022, Kurz and Colleagues [46] found that mice with pancreatic tumors expressed higher levels of CD8 T cells, which are capable of killing cancer cells and/or virus-infected cells, after 30 min of aerobic exercise five days a week. These killer cells express a receptor for IL-15, which is released from muscles during exercise. When CD8 T cells bind to IL-15, they unleash a more potent immune response on pancreatic tumors. This effect prolonged the survival of mice with tumors by about 40 percent, compared to control mice. The findings were confirmed when Kurz et al. [46] analyzed tumor tissue from people with pancreatic cancer. Subjects who performed at least 60 min of aerobic exercise each week produced more CD8 T cells and were twice as likely to survive up to 5 years compared to people in the control group.

The diverse array of exerkines underscores the intricate mechanisms through which exercise influences various physiological processes and provides potential targets for therapeutic interventions in metabolic diseases and other health conditions. Tissue sensitivity and response to exercise vary according to time of day and the alignment of circadian clocks, but the optimal exercise time to elicit a desired metabolic outcome is not fully defined [47]. Exerkines confer adaptive processes between different tissue types, and therefore mediate the preventive and therapeutic effects of physical activity [48]. Exerkines play a critical role in mediating the therapeutic effects of exercise by transmitting molecular signals that promote adaptation, tissue repair, and overall improved health. Understanding the role of exerkines in the response to physical exercise is therefore essential to clarifying the mechanisms through which physical activity benefits health and well-being. It will be necessary and appropriate to study the interindividual variability in exerkines secretion in response to physical exercise. In fact, it is known that some subjects respond with significant changes to the effects of physical exercise [49]. Others, however, do not respond in the same way when subjected to the same exercise (“non-responders”). It is evident that these differences are due to different cellular actions. For example, at the cardiac level, it is the cardiomyocyte that plays a central role both as a target and as an effector of the benefits of physical exercise, but it is certain that other non-cardiomyocyte lineages’ pathways are activated at a systemic level (metabolism, inflammation, microbiome, and aging), resulting in a pleiotropic but personalized effect [50]. Exerkines are known to modulate various physiological processes, including neurocognitive functions, tissue metabolism, and systemic inflammation [43,51]. It has also been shown that exerkines promote neuronal survival, development, and growth, as well as inducing changes in tissue metabolism and signaling [43]. Additionally, exerkines have been associated with anti-inflammatory effects, contributing to the treatment of conditions such as atherosclerosis [51]. The therapeutic potential of anti-inflammatory exerkines in the context of atherosclerosis highlights the diverse roles these molecules play in health and disease [51]. Exerkines are exercise-induced molecules that mediate tissue communication and drive adaptations. Understanding exerkine kinetics and dynamics is crucial for optimizing exercise prescription for disease prevention and treatment and developing exercise-mimicking pharmaceuticals [48].

MicroRNAs (miRNAs), small non-coding RNAs secreted in response to exercise, act as exerkines that regulate gene expression levels [52]. Recent research has highlighted the potential of certain miRNAs to act as exerkines, which are molecules that are released in response to exercise and mediate systemic adaptations and benefits [20]. These exercise-induced circulating miRNAs, termed c-miRNAs, have been proposed to contribute to the multisystemic adaptive effects of physical activity [20]. Specific miRNAs delivered via circulating exosomes can exert protective effects on distal organs, such as the heart, against conditions like myocardial ischemia/reperfusion injury [53]. A recent case–control study of 16 young sedentary men, 16 Olympic endurance athletes, and 16 Olympic endurance athletes, analyzing the miRNA profiles of extracellular vesicles, showed that endurance and resistance athletes had significantly lower levels of miR-16-5p, miR-19a-3p, and miR-451a compared to sedentary people. Interestingly, the miRNA profile observed in extracellular vesicles provided a differential signature of athletes regardless of the type of exercise compared to sedentary people. In fact, miR-25-3p levels were specifically lower in endurance athletes, suggesting an individual and specific response in this group of athletes [54].

miRNAs have been implicated in various biological processes, including angiogenesis, inflammation, and mitochondrial metabolism, making them essential mediators of exercise-induced adaptations [55]. The dynamic regulation of circulating miRNAs during exercise and training underscores their importance in physiological responses to physical activity [55].

There are differences in the secretion of exerkines in response to acute exercise and chronic exercise. Exposure to acute physical exercise is generally associated with responses focused on maintaining metabolic homeostasis, with elevated inflammatory phenomena, while exposure to long-term physical exercise is associated with responses focused on long-term metabolic adaptations and with a decrease in inflammation [56]. In addition, aerobic and anaerobic exercises induce the release of various exerkines. Aerobic exercise is associated with the release of several myokines and exerkines that play significant roles in metabolic regulation (e.g., IL-6, GDF15, FGF21, BDNF, apelin, METRNL), promoting lipolysis and contributing to metabolic regulation and insulin sensitivity [57,58,59]. In contrast, anaerobic exercise primarily stimulates the production of different exerkines, notably lactate, which serves as both an energy substrate and a signaling molecule. Lactate, in fact, can cross the blood–brain barrier and is involved in increasing BDNF expression, thereby promoting neuroplasticity [59,60,61]. Additionally, anaerobic exercise has been linked to the secretion of myokines such as IL-7 and IL-8, which are involved in muscle hypertrophy and anti-inflammatory responses [62]. The combination of anaerobic and aerobic exercises can also enhance the expression of various myokines, indicating that both types of exercise can synergistically improve metabolic health [3]. Notably, the secretion of exerkines such as irisin, which is associated with both aerobic and resistance training, underscores the complex interplay between different exercise modalities in promoting health benefits [63]. Interestingly, a recent study demonstrated that postprandial aerobic exercise regulates tissue-specific triglyceride uptake through angiopoietin-like proteins (ANGPTL3, ANGPTL4, ANGPTL8) [64].

However, there are many other questions: how is this complex network regulated? Is there a lead player? Is it organ-related? There are not many answers to these questions yet, but the MoTrPAC study is beginning to provide some interesting answers. For example, in the small intestine, exercise reduced the expression of some genes associated with inflammatory bowel disease. In the liver, however, it stimulated tissue regeneration. In male rats, eight weeks of resistance training was observed to reduce the amount of specific body fat called subcutaneous white adipose tissue (scWAT). Interestingly, the same amount of exercise did not reduce the amount of scWAT in female rats [14]. Studies have identified specific genes and genetic pathways that are responsive to exercise stimuli and play a role in the production and regulation of exerkines. For instance, research has highlighted the role of myokines, which are induced by exercise and include muscle-derived exerkines, in mediating the beneficial effects of physical activity [58]. Additionally, the identification of genes like *NR4A3*, which respond to exercise-like stimuli and mediate metabolic responses, underscores the genetic basis of the molecular adaptations to exercise [65]. Exercise intensity significantly influences the profile of exerkines secreted by various organs, with distinct patterns emerging based on whether the exercise is low, moderate, or high intensity. High-intensity exercise has been shown to result in the robust release of specific exerkines. For example, the release of IL-6 has been linked to increased glucose uptake and fatty acid oxidation, indicating its role in metabolic regulation during intense exercise [53]. Endurance resistance training induces a different set of exerkines than resistance training, with some studies suggesting that aerobic exercise may lead to greater increases in certain myokines such as irisin and FGF21, which are associated with fat metabolism and energy expenditure [3]. Therefore, understanding secretory dynamics based on exercise intensity may help design exercise programs that maximize the health benefits of physical activity. Hopefully, we will soon understand the molecular basis of these differences, and this will be the first step towards achieving the goal of developing personalized exercise prescription [66].

Overall, the emerging field of exerkines research highlights the importance of these exercise-induced signaling molecules in promoting health, resilience, and disease management. It is precisely this multifaceted role that makes them promising targets for therapeutic interventions. Understanding the molecular mechanisms by which exerkines exert their effects may provide valuable insights into the therapeutic potential of exercise as a non-pharmacological intervention in various health conditions.

## 3. Male Infertility, Physical Exercise, and Exerkines

In recent years, much attention has been paid to bias and how the environment is influencing not only somatic cells but also germ cells due to the importance of epigenetic inheritance across generations also.

Many studies look at factors related to lifestyle and sperm quality. The literature on physical activity and sperm quality is scarce in humans, while it is abundant in research on animal models. In fact, in studies on animals and mice in particular, the positive effect of physical activity on sperm has been widely demonstrated. For example, in mice, paternal preconceptional physical activity induces changes in the expression of sperm miRNAs and DNA methylation associated with obesity and metabolic dysfunction induced by a high-fat diet. Specifically, a preconceptional swimming exercise or dietary intervention (8 weeks) normalized body weight, glucose intolerance, plasma leptin, and C-reactive protein concentrations in obese bulls initially fed a high-fat diet [67]. The researchers also found that a high-fat diet leads to the production of aberrantly expressed X-associated sperm miRNAs involved in cell cycle regulation, apoptosis, and embryo development pathways (miR-503, miR-542-3p, and miR-465b-5p, respectively), which were also restored to control levels by an 8-week diet or exercise intervention. Moderate and regular exercise can be beneficial in male fertility by modulating anti-inflammatory and antioxidative mechanisms. Specific types of exercise, such as resistance training, have been shown to positively affect male factor infertility [67]. Lifestyle interventions, including diet and exercise, can also improve metabolic health and reverse perturbed sperm function in obese individuals, highlighting the importance of a holistic approach to male fertility management [67]. Exercise is important in epigenetic inheritance through generations. In 2015, Denham [68] demonstrated DNA methylation in human sperm following exercise training. Moreover, methylation changes occurred in paternally imprinted genes that are exempt from DNA methylation erasure after fertilization [69,70]. Global sperm DNA methylation was reduced after 3 months of training exercise, thus suggesting that some exercise-responsive genes (CpG sites) are regulated by DNA methylation in somatic and germinal cells. 

In an RCT, a high-intensity training group reported significantly attenuated inflammatory exerkines (IL-6 and TNF-α), oxidative stress (reactive oxygen species and malondialdehyde), and antioxidants (superoxide dismutase, catalase, and total antioxidant capacity) compared to the control (*p* < 0.05), and these changes coincided with favorable improvements in semen parameters, sperm DNA integrity, and pregnancy rate (*p* < 0.05) [71]. A recent systematic review and network meta-analysis, with the aim to evaluate the effectiveness of exercise training on male infertility and seminal markers of inflammation, reported on 2641 fertile and infertile men in seven controlled randomized trials. It appears that moderate-intensity aerobic exercise alone, strength training alone, and the combination of the two significantly improve male infertility [72]. Interestingly, repeated high-intensity interval exercise (HIIE) was found to increase plasma oxytocin (OT) levels in healthy men [73]. Since oxytocin plays a possible role in male reproductive function and infertility [74] through the oxytocin/oxytocin receptor (OT/OTR) system, it is conceivable that this exerkine could have positive effects on the treatment of male infertility.

Recently, mitochondrial tRNAs (mt-tRNAs) and their fragments (mt-tsRNAs) have been identified in human sperm and are therefore susceptible to transmission. mt-tsRNAs in sperm are correlated with body mass index, and paternal overweight at conception doubles the risk of obesity in the offspring as an epigenetic mechanism [75]. Moreover, epididymal spermatozoa, but not developing germ cells, are sensitive to the environment. This study supports the importance of paternal health at conception for offspring metabolism, but, from an evolutionary perspective, these results present a fully reversible mechanism by which male parents influence offspring metabolic risk by transferring mitochondrial signals and thus overcome the mechanisms by which fertilized oocytes eliminate sperm mitochondria. This opens new pathophysiological horizons on the role of paternal offspring transfer of mitochondrial RNAs at fertilization [76]. Even though this was demonstrated regarding metabolic programming, it is highly possible that paternal physical exercise may positively influence sperm-borne mitochondrial RNAs at fertilization. 

## 4. Conclusions

Physical exercise has played and plays a fundamental role in human evolutionary history. Humans have lived a hunter–gatherer lifestyle for tens of thousands of years, hunting and foraging for food, building and maintaining shelter, gathering water, and protecting themselves from predators [77]. This has required walking long distances, and occasionally fighting and escaping from threats. Those with better athletic ability were better equipped to live longer, which favored a selection for exercise as a positive developmental factor [78]. On this basis, the Greeks thought of a competitive quadrennial athletic event called the Olympic Games, which was held in Olympia, Greece, from 776 bce to about 393 ce. It was part of a religious festival that honored Zeus, and the name Olympics was derived from Mount Olympus, home of the Greek gods [79]. 

The shift to a more active lifestyle has led to changes in the human body: exercise burns energy that would otherwise be stored as fat, which, in excessive quantities, increases the risk of cardiovascular disease, type 2 diabetes, and some tumors. A reduction in adipose tissue is one of the ways to reduce weight in individuals with obesity and is necessary to mitigate negative cardio-metabolic comorbidities in obesity. There are two methods that can effectively reduce adipose tissue, and these include (a) change in diet; (b) change in energy expenditure (e.g., exercise). Both aspects have a positive effect on the health of human beings. But the biological basis of these effects was only hypothesized until 1999, when Klarlund Pedersen and his colleagues analyzed blood samples from runners before and after a marathon and discovered that several cytokines, such as IL-6, were secreted immediately after exercise and remained at elevated levels up to 4 h later [9,10,80]. Only later, Safdar et al. [20] coined the term “exerkine” to identify molecules produced in response to acute and chronic exercise and mediate systemic adaptations to exercise. The recent explosion of multiomics—an approach that combines various biological datasets, such as genomics, transcriptomics, epigenomics, proteomics, and metabolomics—has finally allowed researchers to extend their characterization and thus their spatial and temporal classification. The application of multiomics technologies has provided valuable insights into the molecular mechanisms underlying the effects of exercise on the body. Today, it is possible to analyze hundreds of thousands of biologically active molecules and correlate them to form a molecular pathway that is harmoniously activated during exercise [14]. Interestingly, a recent study demonstrated that microbiome transplants from trained donors can improve skeletal muscle disuse atrophy [81]. The results of this study provide compelling evidence to support the use of an exercise-trained microbiome to treat complex, multifactorial diseases. In this context, the bill 287, signed by Senator Daniela Sbrollini, entitled “Provisions for measures to introduce physical activity as a tool for prevention and therapy within the National Health Service”, signed by all parties in the Italian Senate, appears interesting. The bill, at the center of an important battle to promote the role of physical activity as a driver of health, aims to make physical activity prescribable like a drug by general practitioners, pediatricians, and specialists to encourage its use as a tool for prevention and treatment. Although no country has yet explicitly defined exercise as a drug in the legal sense, many are adopting policies that promote its use as a therapeutic intervention, reflecting a paradigm shift in how physical activity is perceived within health care settings.

## 5. Future Directions

Blood biomarker profiling will be increasingly important in the coming years. Michael Snyder at Stanford [82] recently demonstrated that the analysis of thousands of metabolites, lipids, cytokines, and proteins obtained from 10 μL of blood together with physiological information from wearable sensors is able to provide a real-time dynamic evaluation of reactions to a complex mixture of dietary interventions allowing for the discovery of individualized inflammatory and metabolic responses. The combination of wearable devices and multi-omics microsampling during physical exercise will facilitate the dynamic profiling of sports-related health status. This approach has shown promise in predicting individual responses to exercise with respect to metabolic and cardiorespiratory health [19]. By integrating data from different omics levels, researchers have been able to elucidate the complex interplay between exercise-induced molecules and physiological outcomes, paving the way for personalized exercise interventions and targeted therapies.

## Figures and Tables

**Table 1 biomedicines-12-01975-t001:** Exerkines produced by human tissues.

Exerkine	Tissue/Organs	Molecule Type
** *Autocrine effects* **		
12,13-diHOME (12,13-dihydroxy-9Z-octadecenoic acid)	BAT (Brown Adipose Tissue)	Lipid
Apelin	Muscle	Small Peptide
Adiponectin	WAT (White Adipose Tissue)	Protein
BDNF (brain-derived neurotrophic factor)	Brain, muscle	Protein
FGF21 (Fibroblast growth factor 21)	WAT	Protein
HSP72 (heat shock protein 72)	Muscle	Protein
IL-6 (Interleukin 6)	Muscle	Protein
IL-7	Muscle	Protein
IL-15	Muscle	Protein
Irisin (FNDC5)	Muscle	Glycoprotein
Lactate	Muscle	Organic Molecule
LIF (leukaemia inhibitory factor)	Muscle	Protein
miRNAs	Sperm	RNA
Musclin/Ostreocrin	Muscle, bone, brain	Protein
Myostatin	Muscle	Protein
Nitric oxide	Endothelium	Inorganic Molecule
Reactive oxygen species	Muscle	Inorganic Molecules
SPARC (secreted protein acidic and rich in cysteine)	Muscle	Protein
SDC4 (syndecan 4)	Muscle	Protein
TGFβ1 (transforming growth factor β1)	Muscle	Protein
OXT (Oxytocin)	Brain, adipose, systemic metabolism	Small Peptide
METEORIN-LIKE PROTEIN (METRNL)	Muscle, adipose	Protein
** *Paracrine effects* **		
Adiponectin	Muscle	Protein
Angiopoietin 1	Vasculature	Glycoprotein
Angiopoietin-like proteins (ANGPTL3, ANGPTL4, ANGPTL8)	Liver, adipose	Protein
BAIBA (β-aminoisobutyric acid)	WAT, bone	Organic Molecule
BDNF	Nerves	Protein
Fractalkine	Leukocytes	Protein
FGF21	BAT	Protein
GDF15	Liver	Protein
IL-6	WAT	Protein
IL-7	Bone	Protein
IL-8	Vasculature	Protein
IL-13	Muscle	Protein
IL-15	WAT	Protein
Irisin	Adipose tissue, liver	Glycoprotein
LIF	Muscle	Protein
Musclin	Cartilage	Protein
Myostatin	Bone	Protein
SPARC	Extracellular matrix	Protein
SDC4	Muscle	Protein
TGFβ1	Extracellular matrix	Protein
TGFβ2 (transforming growth factor β2)	Muscle, BAT	Protein
VEGF	Endothelium	Protein
OXT	Brain, adipose, systemic metabolism	Small Peptide
METEORIN-LIKE PROTEIN (METRNL)	Muscle, adipose	Protein

The biological pathways of the exerkines reported in the table can be viewed and analyzed on Reactome (https://reactome.org/, accessed on 4 August 2024).

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
