# Peer review of "‘Exerkines’: A Comprehensive Term for the Factors Produced in Response to Exercise"

_biomedicines, 2024, doi:10.3390/biomedicines12091975_

Round 1

Reviewer 1 Report

Comments and Suggestions for Authors

1.- I consider that section 3 contains repetitive information described in section 2 and that the exerkines responsible for all the functions described in this section are not named. Therefore, I suggest that the information contained in section 3 should be included in section 2

2.-I suggest deleting Figure 1, since Table 1 contains all the information of Figure 1... in Table 1 you should include irisin and all the exerkines described in Figure 1, and the nature of each exercine (peptides, nucleic acids or lipids)

3.- I suggest to include a figure that illustrate the intercomunication of all exercines in order to show how all work together and to highlight the importance of exercise for the production of these exerkines 

4.- Which exerkines are produced by aerobic exercise and which ones by anaerobic exercise?

 5.- Does exercise intensity define the profile of exerckines secreted by the different organs?

Author Response

Comment 1: I consider that section 3 contains repetitive information described in section 2 and that the exerkines responsible for all the functions described in this section are not named. Therefore, I suggest that the information contained in section 3 should be included in section 2:

Response 1: We thank the reviewer for this important suggestion. We have therefore merged the two paragraphs and updated the References on the subject. 

Comment 2: I suggest deleting Figure 1, since Table 1 contains all the information of Figure 1... in Table 1 you should include irisin and all the exerkines described in Figure 1, and the nature of each exercine (peptides, nucleic acids or lipids)..

Response 2: We thank the reviewer for this observation. As suggested, we have eliminated Figure 1, integrated and updated Table 1 by inserting a new column indicating the nature of the reported exerkines. 

Comment 3: I suggest to include a figure that illustrate the intercomunication of all exercines in order to show how all work together and to highlight the importance of exercise for the production of these exerkines

Response 3: We thank the reviewer for this important observation. The complexity and the number of interconnections of each of the exerkines cited does not allow an easy visualization in a Figure. Therefore we have inserted at the bottom of the table, the following sentence: "The biological pathways of the exerkines reported in the Table can be viewed and analyzed on Reactome (https://reactome.org/)". This website is free and easily usable by readers. 

Comment 4: Which exerkines are produced by aerobic exercise and which ones by anaerobic exercise?

Response 4 : We thank the reviewer for this suggestion. We have therefore inserted a new paragraph (see below) with the relevant citations and integrated the table in this regard.

“In addition, aerobic and anaerobic exercises induce the release of various exerkines. Aerobic exercise is associated with the release of several myokines and exerkines that play significant roles in metabolic regulation (e.g. IL-6, GDF15, FGF21, BDNF, apelin) promoting lipolysis, and contributing to metabolic regulation and insulin sensitivity[58-60]. In contrast, anaerobic exercise primarily stimulates the production of different exerkines, notably lactate, which serves as both an energy substrate and a signaling molecule. Lactate in fact, can cross the blood-brain barrier and is involved in increasing BDNF expression, thereby promoting neuroplasticity [60-62]. Additionally, anaerobic exercise has been linked to the secretion of myokines such as IL-7 and IL-8, which are involved in muscle hypertrophy and anti-inflammatory responses [63]. The combination of anaerobic and aerobic exercises can also enhance the expression of various myokines, indicating that both types of exercise can synergistically improve metabolic health[3]. Notably, the secretion of exerkines such as irisin, which is associated with both aerobic and resistance training, underscores the complex interplay between different exercise modalities in promoting health benefits[64]. Interestingly, a recent study demonstrated that postprandial aerobic exercise regulates tissue-specific triglyceride uptake through angiopoietin-like proteins (ANGPTL3, ANGPTL8, ANGPTL4)[65].

Comment 5: Does exercise intensity define the profile of exerkines secreted by the different organs?

Response 5: We thank the reviewer for this suggestion. We have therefore inserted a new paragraph (see below) with the relevant citations and integrated the table in this regard.

“Exercise intensity significantly influences the profile of exerkines secreted by various organs, with distinct patterns emerging based on whether the exercise is low, moderate, or high intensity. High intensity exercise has been shown to result in robust release of specific exerkines. For example, the release of IL-6 has been linked to increased glucose uptake and fatty acid oxidation, indicating its role in metabolic regulation during intense exercise[54]. Therefore, understanding the secretory dynamics based on exercise intensity may help design exercise programs that maximize the health benefits of physical activity. Hopefully, we will soon understand the molecular basis of these differences, and this will be the first step towards achieving the goal of developing personalized exercise prescription".

Reviewer 2 Report

Comments and Suggestions for Authors

Dear all,

I would like to start by thanking you for the opportunity to review this manuscript. Overall, the review is informative, but the readability could be greatly improved with some editing. This review might be better if it focused on exerkines and responses to exercise in humans.

However, some points are listed below:

Abstract

The key findings should be cohesively listed at the end of the abstract. Kindly revise this section to make it more cohesive.

Introduction

Lines 32-36, Lines 37-40, Lines 44-54, and 56-58: need to be supported by references. Please go through all the manuscript and try to support each idea that needs to be affirmed/ proven.  

Manuscript body

Line 85: Table 1. Exerkines produced by human tissues: the table is somewhat long and could be presented in a reduced way.

Please, could you explain how the literature review was performed?

Line 190-251: 3. Exercise-induced exerkines: a potent medicine: it is quite a long section, having more than one idea, I advise, if possible, to be divided into two sections: one for the therapeutic effects of physical activity and other for potential of exerkines in promoting health issues.

References

I found some Refs that need to be Abbreviated (e.g., Ref-5 Nature Reviews Endocrinology, Ref-6 The Journal of Physical Fitness and Sports Medicine, and Ref-7 Journal of Neurology and Neuromedicine), please revise all references section.

Please use the ACS style guide to be compatible with Biomedicines’ guidelines. The ACS style guide is recommended, please follow.

Best regards,

Author Response

Comment 1 The key findings should be cohesively listed at the end of the abstract. Kindly revise this section to make it more cohesive:

Response 1: We thank the reviewer for this important suggestion. We have therefore rewritten the Abstract and add a conclusion as below reported:

“Major discoveries involving exerkins highlight their diverse functions and health implications, particularly in metabolic regulation, neuroprotection, and muscle adaptation. These molecules, including peptides, nucleic acids, lipids, and microRNAs, act through paracrine, endocrine, and autocrine pathways to exert their effects on various organs and tissues. Exerkins represent a complex network of signaling molecules that mediate the multiple benefits of exercise. Their roles in metabolic regulation, neuroprotection, and muscle adaptation highlight the importance of physical activity in maintaining health and preventing disease.”

Comment 2: Lines 32-36, Lines 37-40, Lines 44-54, and 56-58: need to be supported by references. Please go through all the manuscript and try to support each idea that needs to be affirmed/ proven.

Response 2: We thank the reviewer for this observation. We provided a modified version iof the Table 1 as also suggested by Rew. 1

Comment 3: Line 190-251: 3. Exercise-induced exerkines: a potent medicine: it is quite a long section, having more than one idea, I advise, if possible, to be divided into two sections: one for the therapeutic effects of physical activity and other for potential of exerkines in promoting health issues.

Response 3: This section was completely rewritten according also to the Rev 1.

Comment 4: Please, could you explain how the literature review was performed?

Response 4: This is a narrative review, with the aim of describing the current state of research on the topic and offering an analysis of the literature reviewed. We have tried to group the strengths and current knowledge of the topic by research categories. In the conclusions we have identified future developments and trends in research in the field.

Comment 5: References

Response 4: Thanks for the comment. We used EndNote 20.6 version and the MDPI style as suggested by Publishers

Reviewer 3 Report

Comments and Suggestions for Authors

The article provides a comprehensive review of the concept of "exerkines" and how these exercise-induced molecules affect various bodily systems. This represents a significant contribution to the fields of sports medicine and exercise biology. I have only minor comments that I believe will improve the paper.

Minor comments

Introduction

Lines 44-54. Please add references.

Conclusion

At the end I think it will be useful to add the conclusion section. 

Future directions

I think, including a "Future Directions" section can help guide the reader on what research or developments could follow based on the findings and discussions in the article.

Author Response

We thank the Reviewer for the positive comments to our manuscript.

Comment 1: Introduction. Lines 44-54. Please add references.

Response 1: Appropriate references as suggested have been added

Comment 2: At the end I think it will be useful to add the conclusion section.

Response 2: A Conclusion section was added as requested  

Comment 3. Future directions. I think, including a "Future Directions" section can help guide the reader on what research or developments could follow based on the findings and discussions in the article.

Response 3: A Future Directions section was added as suggested